# Transcriptome Analysis Reveals Coexpression Networks and Hub Genes Involved in Papillae Development in *Lilium auratum*

**DOI:** 10.3390/ijms25042436

**Published:** 2024-02-19

**Authors:** Yuntao Zhu, Jie Yang, Xiaolin Liu, Tingting Sun, Yiran Zhao, Fayun Xiang, Feng Chen, Hengbin He

**Affiliations:** 1Beijing Key Laboratory of Ornamental Plants Germplasm Innovation and Molecular Breeding, National Engineering Research Center for Floriculture, Beijing Laboratory of Urban and Rural Ecological Environment, School of Landscape Architecture, Beijing Forestry University, Beijing 100083, China; yuntao_zhu@bjfu.edu.cn (Y.Z.); jieyang@hbaas.com (J.Y.); m18510020233@163.com (X.L.); omitto@163.com (T.S.); yiranzhao@bjfu.edu.cn (Y.Z.); 2Industrial Crops Institute, Hubei Academy of Agricultural Sciences, Wuhan 430064, China; xfy323@hbaas.ac.cn (F.X.); chenfeng502wh@163.com (F.C.)

**Keywords:** *Lilium auratum*, petal papillae, anthocyanin accumulation, trichome development, MYB-bHLH-WD40 complex

## Abstract

*Lilium* is a genus of important ornamental plants with many colouring pattern variations. *Lilium auratum* is the parent of Oriental hybrid lilies. A typical feature of *L. auratum* is the presence of red-orange special raised spots named papillae on the interior tepals. Unlike the usual raised spots, the papillae are slightly rounded or connected into sheets and usually have hairy tips. To elucidate the potential genes regulating papillae development in *L. auratum*, we performed high-throughput sequencing of its tepals at different stages. Genes involved in the flavonoid biosynthesis pathway were significantly enriched during the colouration of the papillae, and *CHS*, *F3H*, *F3′H*, *FLS*, *DFR*, *ANS*, and *UFGT* were significantly upregulated. To identify the key genes involved in the papillae development of *L. auratum*, we performed weighted gene coexpression network analysis (WGCNA) and further analysed four modules. In total, 51, 24, 1, and 6 hub genes were identified in four WGCNA modules, MEbrown, MEyellow, MEpurple, and MEred, respectively. Then, the coexpression networks were constructed, and important genes involved in trichome development and coexpressed with anthocyanin biosynthesis genes, such as *TT8*, *TTG1*, and *GEM*, were identified. These results indicated that the papillae are essentially trichomes that accumulate anthocyanins. Finally, we randomly selected 12 hub genes for qRT-PCR analysis to verify the accuracy of our RNA-Seq analysis. Our results provide new insights into the papillae development in *L. auratum* flowers.

## 1. Introduction

Lilies (*Lilium* spp.) are among the world’s most favourable and popular bulbous ornamental crops and are widely spread throughout the Northern Hemisphere. Lilies, which are usually cut flowers and garden plants, are considered to have great ornamental value and commercial value due to their various flower shapes, colours, and fragrances, and this has sparked strong interest in cultivating lilies with novel flower characteristics. The Royal Horticulture Society in the UK divides lily varieties into nine hybrids according to their characteristics. Oriental hybrid lilies are among the most important and horticulturally used lily hybrids. *L. auratum*, which is indigenous to Japan’s Honshu, Shikoku, Kyushu, and Ryukyu Islands, has contributed to the production of Oriental hybrid lilies [1]. A typical feature of *L. auratum* flowers is that their white interior tepals have gold–yellow veins radiating from the centre and are densely covered with red-orange special raised spots named papillae. In contrast to the usual raised spots, these papillae are slightly rounded or connected into sheets and usually have hairy tips [2]. The papillae of *L. auratum* and several Oriental hybrid lilies have unique ornamental properties.

Colour is a visible trait of plants and is highly important to their ornamental value. The colour formation of flowers relies on the types and contents of their pigments. In wild lilies and lily cultivars, pink and red-purple colours predominate as a result of anthocyanin accumulation [3]. Anthocyanins, which are responsible for plant colouration, are a class of flavonoids widely found in plants [4,5]. In addition to colour variation, lily flowers often develop red-purple raised spots on their interior tepals with the accumulation of anthocyanin pigments [3,6]. Anthocyanin accumulation is a complex process involving a series of anthocyanin biosynthetic enzyme activities, which have been well studied [7,8]. The anthocyanin biosynthesis pathway is a branch of the flavonoid biosynthesis pathway [9,10]. Phenylalanine ammonia- = lyase (PAL), chalcone synthase (CHS), chalcone isomerase (CHI), flavanone 3-hydroxylase (F3H), flavonoid 3′-hydroxylase (F3′H), dihydroflavonol 4-reductase (DFR), anthocyanin synthase (ANS), anthocyanidin reductase (ANR), and flavonoid-3-O-glucosyltransferase (UFGT) are key enzymes in the anthocyanin biosynthesis pathway [11]. The transcription level of anthocyanin biosynthesis enzymes primarily determines their activity, which is regulated by interactions between the R2R3-MYB and bHLH (basic helix–loop–helix) transcription factors. The metabolism of flavonoids, polyphenolic metabolites, is regulated by the MYBs, bHLH, and WD40 proteins via the formation of a well-described MYB-bHLH-WD40 complex [12,13,14].

Papillae are raised spots similar to trichomes. Trichomes occur in a wide variety of plant species and play a vital role in their development [15,16]. Trichomes develop from the epidermal cells and are specialised structures that grow out of plant epidermal tissues. These trichomes are widely distributed on the surface of different organs or tissues in different plants and exhibit various morphologies [16,17]. Along with the stomata, cutin, and wax on the epidermis, trichomes not only play key protective roles through synthesising, storing, and secreting many important substances, but their secondary metabolites also have wide applications and high economic value [18,19,20]. For example, cotton (*Gossypium hirsutum*) trichomes maintain the correct flower bud shape, which is essential for cotton seed production [21]. Previous studies have shown that the development of plant trichomes is regulated by a complex molecular network induced by phytohormones, as well as environmental factors. The positive regulators in the molecular network consisted of a MYB-bHLH-WD40 complex, which included one WD40 family protein, TTG1 (TRANSPARENT TESTA GLABRA1); four bHLH-like transcription factors, *GL3* (*GLABRA3*), *EGL3* (*ENHANCER OF GLABRA3*), *TT8* (*TRANSPARENT TESTA 8*), and *MYC1*; and three R2R3 MYB-related transcription factors, *GL1* (*GLABRA1*), *MYB23*, and *MYB5* [22,23,24,25,26,27,28,29,30]. These functionally redundant genes form a MYB-bHLH-TTG complex that binds to the promoter of *GL2* (*GLABRA2*) [31].

The previous research has suggested that anthocyanin accumulation and trichome development may be jointly regulated by the same genes. *TTG1* plays a key role in both regulating the development of trichomes and the accumulation of anthocyanin [22,32]. JAZ (JASMONATE ZIM-DOMAIN) proteins, known as inhibitors of the JA (jasmonic acid) signalling pathway, interact with the bHLH (*TT8*, *GL3*, *EGL3*) and R2R3 MYB transcription factors (*MYB75* and *GL1*), which are essential components of MYB-bHLH-WD40 complexes, to repress JA-regulated anthocyanin accumulation and trichome initiation. *TT8* allows strong and cell-specific flavonoid accumulation through feedback regulation involving *TTG1*, homologous *MYB*, and *bHLH* transcription factors, which is consistent with the known involvement of the *TT8* bHLH factor in proanthocyanin, anthocyanin, and mucilage biosynthesis [33]. The regulatory mechanisms of anthocyanin accumulation and trichome development are regulated by the MYB-bHLH-WD40 complex, which contains some of the same genes [25,27,33].

Understanding the molecular mechanism of papillae development in *L. auratum*, which is one of the parents of the major cultivars of Oriental hybrid lilies, is crucial for lily breeding. Here, we provide new insights into the regulatory mechanism of papillae development in *L. auratum* using high-throughput sequencing. Transcriptomic analysis of the differentially expressed genes (DEGs) revealed that a subset of anthocyanin biosynthesis genes was upregulated during the colouration of the papillae, which might result in colour variation in the papillae. We conducted data mining based on the pattern of papillae development using weighted gene coexpression network analysis (WGCNA) and identified several hub genes that may be vital to this process. Finally, we constructed coexpression regulatory networks related to papillae development. Our results provide new insights into the identification of the functional genes involved in papillae formation and lay the foundation for ornamental trait improvement via the use of advanced breeding technology in lily breeding.

## 2. Results

### 2.1. RNA-Seq De Novo Assembly and Transcriptome Analysis

To understand the molecular mechanism of papillae development in the *L. auratum* inner perianth, fifteen strand-specific cDNA libraries from S1, S2, S3, S3S, and S3 W (Figure 1a; for details, see Materials and Methods) were sequenced, including three biological replicates for each stage. A total of 105.6 Gb of clean data were obtained, and the amount of valid data for each sample was distributed between 6.69 and 7.34 Gb. The Q30 bases were distributed between 86.47 and 91.24%, and the average GC content was 49.33%. After de novo assembly and redundancy, 39,013 unigenes were obtained. The total length, average length, and N50 length were 44,899,324 bp, 1151 bp, and 1662 bp, respectively. Benchmarking Universal Single-Copy Orthologues (BUSCO) assessment found that 78.47%, 4.24%, 5.07%, and 12.22% of the total 1440 conserved single-copy orthologues were annotated as complete and single-copy (C,S), complete and duplicated (C,D), fragmented (F), and missing (M), respectively (Table 1). The biological functions of the unigenes were annotated using the Nr (61.33%), Swiss-Prot (46.26%), KEGG (13.38%), GO (40.70%), and Pfam (42.15%) databases.

To find the correlation between the anthocyanin synthesis pathway and *L. auratum* papillae development, we identified DEGs between different stages (Figure 1b). A total of 5702 DEGs were identified between S2 and S1; a total of 2294 genes were upregulated, and 3408 genes were downregulated. We performed KEGG enrichment analysis on the DEGs between S2 and S1 (Figure 1c), and the most enriched pathway was flavonoid biosynthesis (ko000941). We then conducted Gene Set Enrichment Analysis (GSEA) between S2 and S1 using genes annotating the flavonoid biosynthesis pathway, and the results showed that the genes in this pathway were significantly upregulated (enrichment score = 0.70, *p* value < 10^−5^) in S2 (Figure 1c). A total of 1488 DEGs were identified between S3S and S3 W, with 1101 genes upregulated and 387 genes downregulated (Figure 1d). KEGG enrichment analysis of the DEGs between S3S and S3 W revealed that the flavonoid biosynthesis pathway was also enriched. The GSEA results (Figure 1d) showed that the genes annotating the flavonoid biosynthesis pathway were significantly upregulated in S3S (enrichment score = 0.42, *p* value < 10^−5^).

### 2.2. Enzyme-Encoding Genes Involved in Anthocyanin Biosynthesis

According to the NR, Swiss-Prot, and KEGG database annotations and alignments of the *L. auratum* transcriptome using orthologous genes from *Arabidopsis*, we identified 21 candidate transcripts that encode 10 key enzymes involved in anthocyanin biosynthesis (Appendix A and Figure 2). These transcripts are differentially expressed. The normalised expression abundance is shown on the heatmap in Figure 2. The relatedness of the gene expression was shown using a k-means clustering heatmap (Appendix A). The transcript levels of *CHS*, *F3H*, *F3′H*, *FLS*, *DFR*, *ANS*, and *UFGT* were significantly upregulated between S2 and S1. These genes are known to contribute to anthocyanin accumulation [7]; the upregulation was consistent with the phenotypic change in S2 when papillae colouration began (Figure 1a). *ANR* was highly expressed in S1 but had low expression in S2 and thereafter, which is consistent with the knowledge that ANR has a negative effect on anthocyanin accumulation [7].

### 2.3. Coexpression Network Analysis Reveals Key Gene Sets in Papillae Development

WGCNA was performed to construct coexpression networks and to reveal the functions of networks instead of individual genes. A total of 14 distinct modules were obtained, and one named MEgrey was reserved for genes outside of all modules (Figure 3 and Appendix A). Four modules with specific expression patterns (MEbrown, 1940 genes; MEyellow, 1413 genes; MEpurple, 270 genes; and MEred, 433 genes) were further analysed (Figure 3d). The relatedness of the gene expression in these four modules was shown using k-means clustering heatmaps, respectively (Appendix A). Some genes involved in the anthocyanin biosynthesis pathway, including *CHS*, *F3H*, *F3′H*, *DFR*, *ANS*, and *UFGT,* were found in MEbrown. The four gene sets of the MEbrown, MEyellow, MEpurple, and MEred modules were subjected to KEGG pathway enrichment analysis (Appendix A). The eigengenes in the MEbrown, MEpurple, and MEred modules were significantly enriched in the flavonoid biosynthesis pathway, and the eigengenes in the MEbrown module were also enriched in the anthocyanin biosynthesis pathway (*p* value < 0.05). In the MEyellow and MEred modules, genes were significantly differentially expressed between S3S and S3 W, which may indicate the regulation of papillae development.

### 2.4. Identification of Hub Genes

To further identify the key regulatory genes in these modules, degree values were calculated to show the number of connections (edges) to each gene in each module. We identified 51, 24, 1, and 6 hub genes in MEbrown, MEyellow, MEpurple, and MEred, respectively (Appendix A). In the MEbrown cohort, the hub gene with the highest degree value (1787 in 1940 genes) was *TT8*. In addition to the enzyme-encoded genes involved in anthocyanin biosynthesis, several genes related to trichome formation or lateral root initiation, such as *KINESIN-13A*, *GEML8*, and *AUX1*, were identified. Several genes that both control anthocyanin accumulation and trichome development were identified, e.g., *TT8* and *TTG1*. A total of eight bHLH family transcription factors, two MYB family transcription factors, and two WD40 repeat protein-encoding genes (*RACK1B* and *RUP2*) were identified. Several plant hormone-related genes, e.g., *GAI*, *PIN4*, and *IAA8*, were identified as hub genes. In MEyellow, the hub gene with the highest degree value (1363 of 1413 genes) was *BBX19*. Interestingly, we identified three *JAZ-LIKE* genes in MEyellow. *GEM* (*GL2-EXPRESSION MODULATOR*) and an R2R3-MYB involved in root and hypocotyl epidermal cell fate determination (*MYB66*) were identified and may be related to the regulation of papillae development. In MEpurple, the only hub gene was *MYBC2*, an R2R3-MYB. In MEred, the top hub gene was *NCED4*, which is related to the carotenoid biosynthesis pathway. A positive regulator of the hair cell differentiation gene *CPC*, an R3-MYB, was identified. These results indicated that genes related to the anthocyanin biosynthesis pathway, carotenoid biosynthesis pathway, and trichome development may play vital roles in papillae development.

### 2.5. Coexpression Network Visualisation

There were 1940, 1413, 270, and 433 eigengenes in MEbrown, MEyellow, MEpurple, and MEred, respectively. To explore how the hub genes were coexpressed with other eigengenes in each module, we used Cytoscape to visualise the coexpression network of these four modules (Figure 4 and Appendix A). Hub genes are represented as coloured nodes in the inner circle, and other eigengenes are represented in the outer circle. The coexpression relationships are represented by coloured edges; a thicker edge and darker colour indicate stronger coexpression (Appendix A). To further explore the coexpression relationships among these hub genes, we additionally visualised the coexpression network between the hub genes in MEbrown, MEyellow, and MEred (Figure 4). In MEbrown, *RACK1B*, which encodes a WD40 domain-containing protein, had the most, strongest coexpression relationships with the other hub genes (Figure 4a). *ELF4-LIKE 4* (*EARLY FLOWERING 4 LIKE 4*), which is a part of the autonomous pathway complex, had the most and strongest coexpression relationships with the other hub genes in MEyellow (Figure 4b). In MEred, the hub gene that had the most and strongest coexpression relationships with the others was *NCED4* (Figure 4c). There was only one hub gene in MEpurple, and the coexpression network of MEpurple is shown in Appendix A. The expression profiles of all these hub genes are represented using a heatmap (Figure 5).

### 2.6. Expression Profiles of Trichome Development Regulatory Genes

The WGCNA results showed that some trichome development regulatory genes were coexpressed with anthocyanin biosynthesis enzymes, and some were found in MEyellow with significantly high expression levels during the stage when the papillae formed. These results suggest that trichome-related genes may also be related to lily papillae development. Therefore, we analysed the expression patterns of these genes using sequence alignment with homologous sequences from *Arabidopsis* (Table 2 and Appendix A). The sequence alignment showed that positive regulators of the trichomes *GL1* and *MYB23* both targeted TRINITY_DN13618_c0_g2_i2_2, the hub gene of MEpurple, in our transcriptome and that this transcript was highly homologous to *MYB66*. Moreover, the transcript in MEbrown TRINITY_DN21331_c0_g1_i1_1 was highly homologous to *MYC1*, *EGL3*, and *TT8*. Analyses of the expression profiles showed that most of the trichome-related genes were highly expressed in S1 or S2, and the positive regulators *GL1*, *MYB23*, *BRICK1*, and *TEM* were more highly expressed in S3 and S3S than in S3 W (Appendix A). The results indicated that the hub gene of MEpurple, TRINITY_DN13618_c0_g2_i2_2, an R2R3-MYB, may play a highly important role in papillae development.

### 2.7. qRT-PCR Validation of the Hub Genes

qRT-PCR analyses were performed to verify the gene expression patterns obtained via RNA-seq, and the interior tepals of *L. auratum* at three developmental stages (S1–S3) were used as samples for RNA extraction and cDNA synthesis. β-actin was used as the internal control. We randomly selected twelve genes from our identified hub genes for qRT-PCR validation. Overall, most of the gene expression patterns measured using qRT-PCR were consistent with those obtained in the RNA-seq analysis (Figure 6), indicating the reliability and accuracy of our RNA-seq analysis. To further explore whether these hub genes may be involved in imparting papillae formation in other lilies, we identified these twelve hub genes in the *L. speciosum* var. *gloriosoides* transcriptome from lily flowers that were sampled at similar development stages (i.e., S1, S2, and S3), and we presented the gene expression levels (FPKM) as histograms (Appendix A). Most of these hub genes showed similar expression trends in *L. auratum* and *L. speciosum* var. *gloriosoides*.

## 3. Discussion

The main aim of our study was to reveal the transcriptional control of papillae development in *L. auratum*, which includes two aspects: the colouration of the spots and the rise of the spots. Previous studies have shown that the colouration of red-purple raised spots on the interior tepals is an attribute of the accumulation of anthocyanin pigments [3,6]. Our results are consistent with this statement. The results of the DEG analyses showed that genes involved in the flavonoid biosynthesis pathway were significantly upregulated in S2 compared to those in S1 and in S3S compared to those in S3 W. Many key enzymes involved in anthocyanin biosynthesis were highly expressed in S2, which is the stage in which the papillae changed from clear to dark purple. In a recent study, seven key structural genes, namely *CHS*, *CHI*, *F3H*, *F3′H*, *DFR*, and *ANS*, were shown to be activated during anthocyanidin biosynthesis in *L. leichtlinii* var. *maximowiczii* spot colouration [34]. In our study, *CHS*, *F3H*, *F3′H*, *FLS*, *DFR*, *ANS*, and *UFGT* were significantly activated during papillae colouration, which is consistent with the findings of previous studies. In purple-red chokecherry leaves (*Padus virginiana*), the expression of flavonoid biosynthesis genes (*PAL*, *CHS*, and *CHI*) and their transcriptional regulators increased during purple-red periods [35]. In *Lycium* species with black and red fruit, the expression levels of *CHS*, *CHI*, *F3′5′H*, *DFR*, *ANS*, and *UFGT* were highly consistent with the accumulation of anthocyanin [36,37]. These studies indicate the importance of *CHS*, *F3H*, *F3′H*, *DFR*, *ANS*, and *UFGT* in anthocyanin biosynthesis. FLS is an enzyme that converts colour flavonoid intermediates into colourless flavanols, and in *Dendrobium* hybrids, high expression levels of FLS result in white flowers [38]. Interestingly, in our study, *FLS* was highly expressed together with other enzymes involved in anthocyanin biosynthesis. This may have resulted in anthocyanin accumulation only in the papillae and not in other parts of the petals.

Afterwards, the chlorophyll content decreases, and the periphery of the petals gradually fades to white, which may be regulated by *NON-YELLOWING 1* (*NYE1*, also known as *STAY GREEN* or *STG*). *NYE1* catalyses a reaction that removes magnesium from chlorophyll, which is the initial step in chlorophyll degradation, and the overexpression of *NYE1* results in the production of jasmonate [39]. In our study, *NYE1* was almost unexpressed in S1 or S2 but was extremely highly expressed in S3/S3S/S3 W. This may illustrate the fading of the lily petals from their original green to white.

Jasmonate (JA) mediates anthocyanin accumulation and trichome initiation, and JAZ proteins are inhibitors of the JA signalling pathway. A previous study showed that JA regulates MYB-bHLH-WD40-complex-mediated anthocyanin accumulation and trichome initiation in a COI1-dependent manner. Degradation of the JAZ proteins abolishes the interactions of the JAZ proteins with the bHLH and MYB factors, subsequently aiding the transcriptional function of the MYB-bHLH-WD40 complex in activating downstream signals [14]. In *Arabidopsis*, JA functions as a preferential lateral root (LR) inhibitor, and the treatment of seedlings with JA suppressed the expression of auxin-inducible genes related to LR formation [40]. JAZ4, a member of the JAZ protein family, functions as a negative regulator of auxin signalling in the root tissues above the apex but functions as a positive regulator of auxin signalling elsewhere [41]. *TCP15*, a member of the TEOSINTE BRANCHED 1-CYCLOIDEA-PCF (TCP) transcription factor family, interacts with many JAZ proteins (e.g., JAZ1, JAZ2, JAZ3, etc.) [42]. *TCP15* is a repressor of auxin biosynthesis, and it is required for elongation and gene expression responses to auxin [43,44]. These studies show that JAZ proteins may be involved in elongation by affecting auxin signalling. In our study, we identified three JAZ proteins as hub genes in MEyellow, and the expression levels of these JAZs were low in the early stages and high in S3. These results indicate that JA-mediated regulation of anthocyanin accumulation and trichome initiation may occur in the early stage, and JAZs may be involved in trichome initiation by affecting auxin signalling.

To explore the nature of the increase in papillae and the associated genes, we performed a coexpression analysis via WGCNA. We constructed a coexpression network and analysed four modules. A total of 82 hub genes were identified, and the coexpression networks were visualised. Some trichome-related genes were found to be coexpressed with anthocyanin biosynthesis genes, and some genes that regulate both anthocyanin accumulation and trichome development were found. We subsequently analysed important genes involved in trichome development and identified two transcripts in our transcriptome that may be extremely important, i.e., TRINITY_DN13618_c0_g2_i2_2 (R2R3-MYB) and TRINITY_DN21331_c0_g1_i1_1 (bHLH transcription factor). Many important genes are homologous to these two transcripts. These genes, together with TTG1, a WD40 repeat protein, may form an MYB-bHLH-WD40 complex that regulates anthocyanin accumulation and trichome development. This conjecture needs to be confirmed using further in vivo and in vitro experiments. These results suggest that papillae, which are raised and coloured spots, may be essentially trichomes. In our study, we also identified other potential members of the MYB-bHLH-WD40 complex and several potential regulators, providing a basis for further research on the mechanism of *L. auratum* papillae formation.

## 4. Materials and Methods

### 4.1. Plant Materials and Sampling

The *L. auratum* plants were planted under greenhouse conditions at Beijing Forestry University, Haidian District, Beijing, China (Beijing Haidian District N 39°95′ E 116°30′). The flower tissues were collected according to the colouring of the tepals at different stages of flower bud development. In the first stage (S1), the interior tepals of *L. auratum* were green and 5.5 cm long, and uncoloured papillae could be observed (Figure 1a). In the second stage (S2), the interior tepals were light yellow-green and 7.5 cm long, and the papillae were coloured purple (Figure 1a). In the third stage (S3), the flowers were in full bloom, the interior tepals were 13 cm long, and the papillae turned red-orange from purple (Figure 1a). In S1, S2, and S3, the whole tepals were collected and labelled S1, S2, and S3, respectively. The outer white area of the inner perianth in S3 was sampled and labelled S3 W. The coloured papillae of the inner perianth in S3 were detached and sampled and labelled S3S. A total of five samples were snap-frozen in liquid nitrogen and kept at −80 °C for subsequent experiments.

### 4.2. RNA-Seq De Novo Assembly and Transcriptome Analysis

RNA sequencing and de novo assembly were subsequently conducted by OE Biotech Co., Ltd. (Shanghai, China). Five samples with three replicates per sample were subjected to high-throughput Illumina sequencing. Tissue collection and preparation for RNA-seq were performed as previously described. Total RNA was extracted using the mirVana miRNA Isolation Kit (Ambion, Austin, TX, USA) following the manufacturer’s protocol. The libraries were constructed using the TruSeq Stranded mRNA LTSample Prep Kit (Illumina, San Diego, CA, USA) according to the manufacturer’s instructions. Then, these libraries were sequenced on the Illumina sequencing platform (HiSeq^TM^ 2500, Illumina, San Diego, CA, USA), and 125 bp paired-end reads were generated. The raw data generated by Illumina were processed using Trimmomatic (version 0.36) to obtain clean data [45]. Low-quality reads, reads containing poly-N, and adaptors were removed. The clean reads were assembled into expressed sequence tag clusters (contigs) and de novo assembled into transcripts using Trinity (version 2.4) via the paired-end method based on the de Bruijn graph algorithm [46]. The longest transcript was chosen as a unigene based on the similarity and length of a sequence determined using CD-HIT (version 4.6) with the default settings for subsequent analysis [47]. BUSCO assessment was performed to estimate the transcriptome completeness and redundancy [48].

The functions of the unigenes were annotated via alignment with the NCBI non-redundant (NR), Pfam, and Swiss-Prot databases using BLASTx with a threshold E-value of 10^−5^ [49]. The proteins with the highest hits to the unigenes were used to assign functional annotations. Gene Ontology (GO) classification was performed by mapping the relationships between the Swiss-Prot and GO terms based on the Swiss-Prot annotation. Kyoto Encyclopedia of Genes and Genomes (KEGG) annotations were assigned by mapping the unigenes to the KEGG database. The read counts and fragments per kilobase of transcript per million mapped reads (FPKM) values of the unigenes were calculated using Bowtie2 (version 2.3.3.1) and eXpress (version 1.5.1) [50,51]. DEGs were identified using the DESeq functions “estimateSizeFactors” and “nbinomTest” with a threshold *p* value < 0.05 and |log_2_FoldChange| (FC) > 1 for significantly differential expression [52].

### 4.3. Coexpression Network Construction with WGCNA

WGCNA was performed using the R (version 4.3.2) WGCNA package (version 1.7.2-1) [53]. To improve the data availability, unigenes whose gene expression levels were too low (the sum of the gene expression was lower than 10^−5^) or whose expression was zero in more than 50% of the samples were filtered out, after which the remaining 25% of the unigenes, whose gene expression changes were not obvious enough, were filtered out using the median absolute deviation (MAD) method. To construct the scale-free network, “R^2^ = 0.9” was set, after which the soft threshold power was selected. The coexpression network and WGCNA modules were constructed by computing the topological overlap matrix (TOM) with “power” = 8, “minModuleSize” = 50, “mergeCutHeight” = 0.2, and “TOMType” = “unsigned”. The module eigengene adjacency heatmap and the TOM heatmap were generated using the WGCNA package.

### 4.4. Identification of the Hub Genes from WGCNA

The number of genes that were coexpressed with each eigengene was calculated to identify hub genes using the Cytoscape (version 3.10.1) “Analyse Network” tool [54]. Genes with a high degree value (more than 1/3 of the eigengenes in each module) and potential related functional annotation were identified as hub genes of the corresponding WGCNA module. The key modules obtained from the WGCNA were visualised using Cytoscape (version 3.10.1). Coexpression relationships with a “weight” ≥ 0.15 obtained from WGCNA were used for calculating the degree and coexpression network visualisation of the key modules.

### 4.5. Quantitative Real-Time PCR (qRT-PCR) and Statistical Analysis

The expression levels of the selected hub genes were validated using quantitative real-time PCR (qRT–PCR). Specific primers (Appendix A) were designed using Primer3 (https://primer3.ut.ee/, accessed on 12 December 2023). cDNAs were synthesised using ReverTra Ace^®^ qPCR RT Master Mix with gDNA Remover (TOYOBO, Osaka, Japan). TB Green^®^ Premix Ex Taq™ II (Tli RNaseH Plus) (Takara, Kyoto, Japan) was used for the detection of qRT-PCR products using a CFX384 Real-Time PCR Detection System (Bio-Rad, Hercules, CA, USA). The RT-qPCR program was initiated with a denaturation step (94 °C for 5 min), followed by 40 cycles of PCR (98 °C for 10 s, 53 °C for 30 s, 68 °C for 90 s) and extension (68 °C, 6 min). Three independent RNA samples from each stage were used as three biological replicates to ensure reproducibility, and three technical replicates were performed for each sample to ensure reliability. The relative gene expression levels of the hub genes were determined using qRT-PCR and using β-actin as the internal control. The 2^−∆∆Ct^ method was used to compute the relative gene expression [55].

## 5. Conclusions

In this study, transcriptome analyses were used to identify the hub genes responsible for papillae development on the petals of *L. auratum*. A total of 82 hub genes were identified in four distinct WGCNA modules. The transcripts of *CHS*, *F3H*, *F3′H*, *FLS*, *DFR*, *ANS*, and *UFGT* were significantly upregulated during the colouration of the papillae. WGCNA revealed that several trichome development regulatory genes, including *KINESIN-13A*, *AUX1*, *LRL1*, *GEML8*, and *IAA8*, were coexpressed with these anthocyanin accumulation genes. Additionally, a group of genes that regulate trichome development, including *JAZ1*, *GEM*, *MYB66*, *PFA1*, and *ABIL3*, was found in a module with significantly high expression levels during the stage when the papillae formed. These results indicated that the increase in spots may be regulated by trichome regulatory genes and that the papillae on the petals may essentially be trichomes. The results of this study provide valuable information and new insights for further study of the mechanism of development of lily papillae.

## Figures and Tables

**Figure 1 ijms-25-02436-f001:**
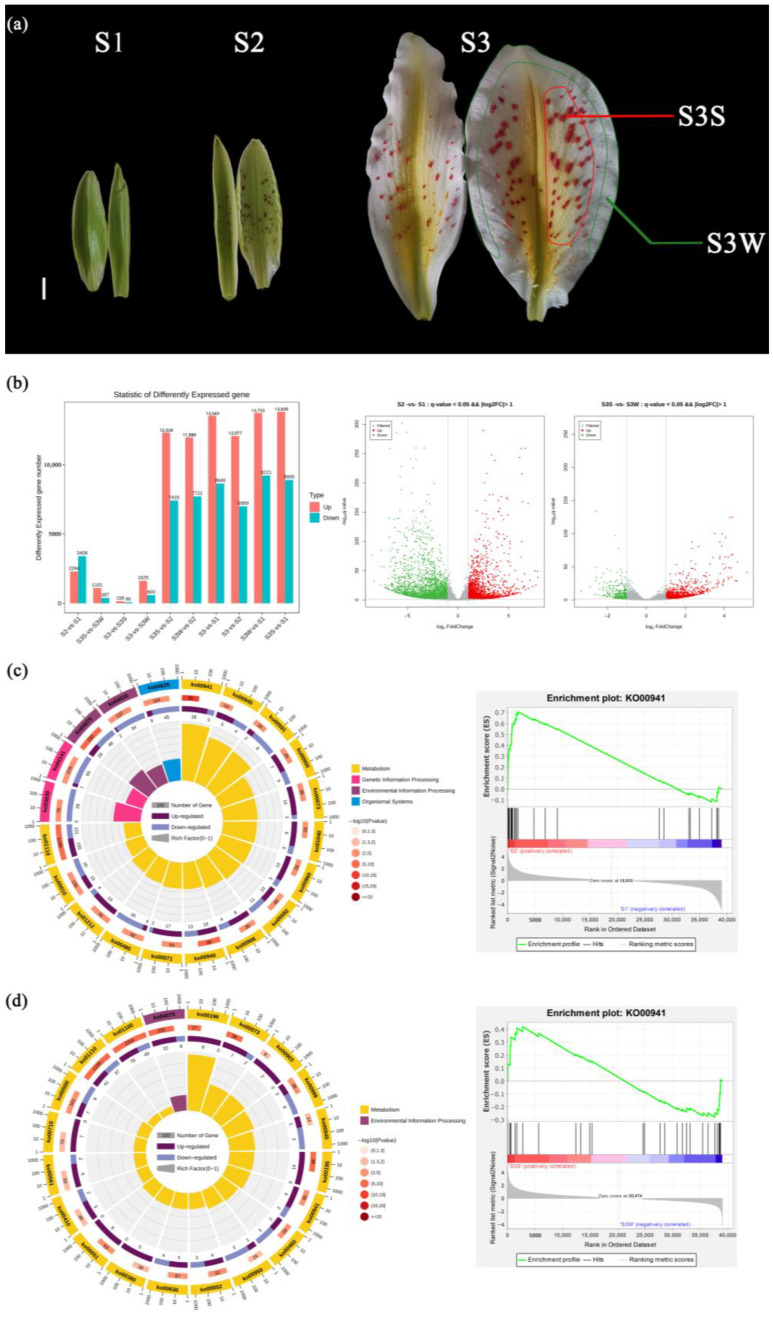
Developmental stages of *L. auratum*. and identification and KEGG/GSEA enrichment analysis of DEGs. (**a**) In S1, the interior tepals were 5.5 cm long, and uncoloured papillae could be observed. In S2, the interior tepals were 7.5 cm long, and the papillae turned purple. In S3, the interior tepals were 13 cm long, and the papillae turned red-orange. The outer white area of the tepal was sampled and named S3 W (marked with a green line). The red-orange papillae were detached and sampled and named S3S (marked with a red line). For details on the sampling, see Materials and Methods. The scale bar (white line) represents 1 cm. (**b**) DEGs were identified with a threshold *p* value < 0.05 and |log_2_FoldChange| (FC) > 1. (**c**) KEGG enrichment analysis (left) and GSEA enrichment analysis (right) of DEGs between S2 and S1. The top 20 enriched KEGG pathways were used for the KEGG enrichment circle diagram (from the outside to the inside, the first circle represents the ko-id of the pathway; the second circle represents the number and *p* value of background genes in this pathway, and the more genes there are, the longer the bar is; the third circle represents the number of DEGs in this pathway, with upregulated genes annotated purple and downregulated genes annotated blue; and the fourth circle represents the value of the Rich factor in this pathway). (**d**) KEGG enrichment analysis (left) and GSEA enrichment analysis (right) of DEGs between S3S and S3 W.

**Figure 2 ijms-25-02436-f002:**
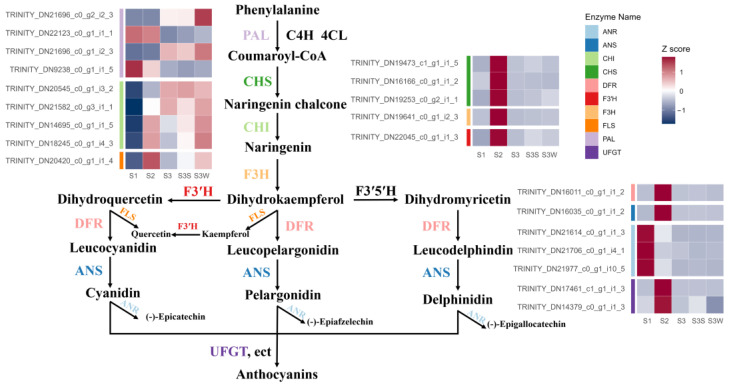
Schematic representation of the anthocyanin biosynthesis pathway in *L. auratum*. The expression abundance of the transcripts was normalised to the Z score and represented on a heatmap. The colour labels to the left of the heatmap indicate that the corresponding transcripts encode enzymes.

**Figure 3 ijms-25-02436-f003:**
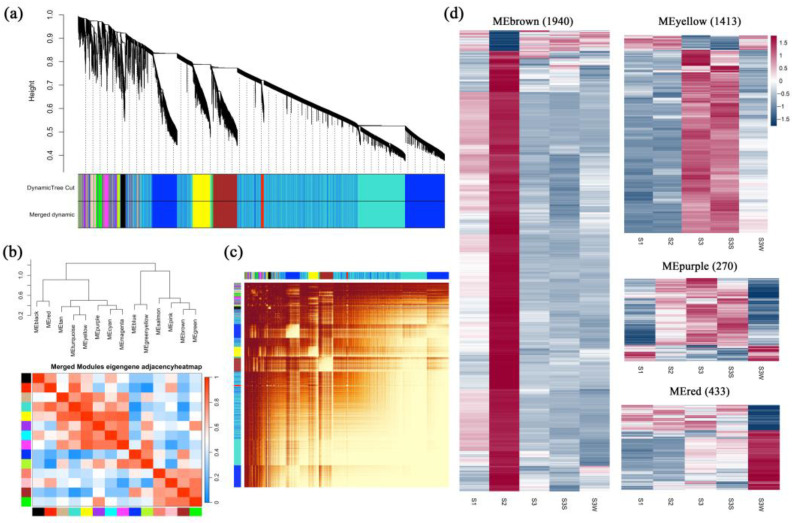
Weighted gene coexpression network analysis (WGCNA). (**a**) Cluster dendrogram. Fourteen distinct modules and a grey module were obtained. (**b**) Eigengene adjacency heatmap of the modules. The cluster tree indicates the correlations between the modules. A heatmap representing the Pearson correlation coefficient is shown. (**c**) Graphical representation of the topological overlap matrix (TOM). This matrix was used to construct the coexpression network. (**d**) Gene expression profiles of four modules that were further analysed are represented using heatmaps. FPKMs were scaled and clustered by row.

**Figure 4 ijms-25-02436-f004:**
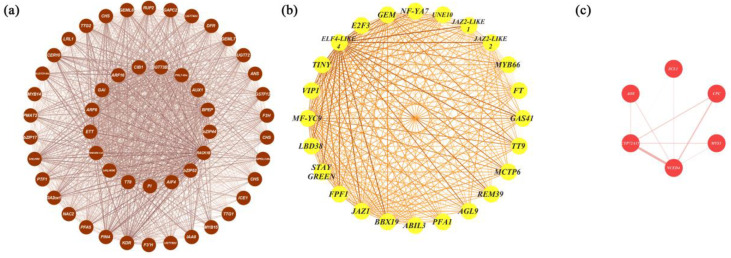
Coexpression network visualisation of hub genes from MEbrown, MEyellow, and MEred. (**a**) The coexpression network of the hub genes identified from the MEbrown cohort. The coexpression network consisted of 51 hub genes identified in MEbrown, ranked by degree, from *TT8* (inner circle) to *KDR* (outer circle) clockwise. (**b**) The coexpression network of the hub genes from MEyellow. The coexpression network consisted of 24 hub genes identified in MEyellow, ranked by their degree value, starting from *BBX19* clockwise. (**c**) The coexpression network of the hub genes from the MEred cohort. The coexpression network consisted of six hub genes identified in MEred, ranked by their degree value, starting from *NCED4* clockwise. The thicker the edge and the darker the colour, the greater the weight, indicating a stronger coexpression relationship.

**Figure 5 ijms-25-02436-f005:**
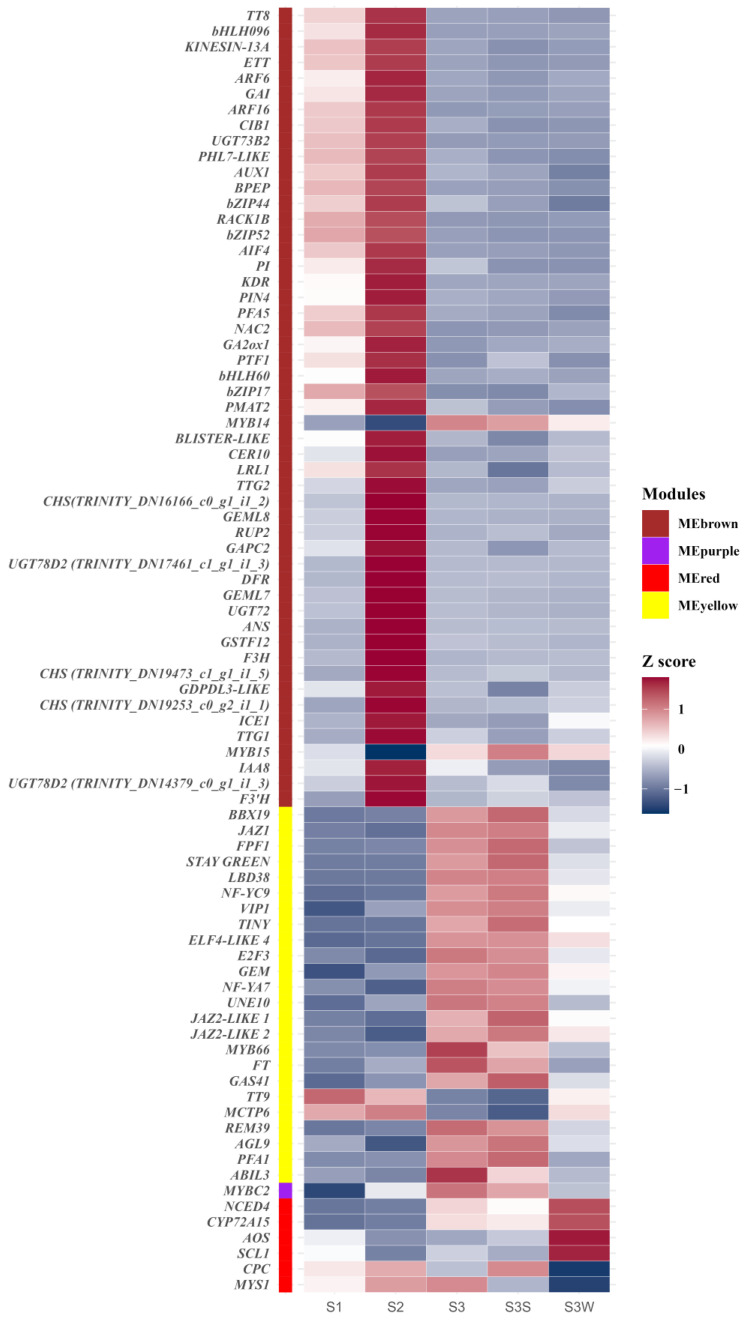
Gene expression profiles of hub genes in MEbrown, MEpurple, MEred, and MEyellow. The colour labels show the module to which the gene belongs. Gene expression was normalised to the Z score and represented using a heatmap.

**Figure 6 ijms-25-02436-f006:**
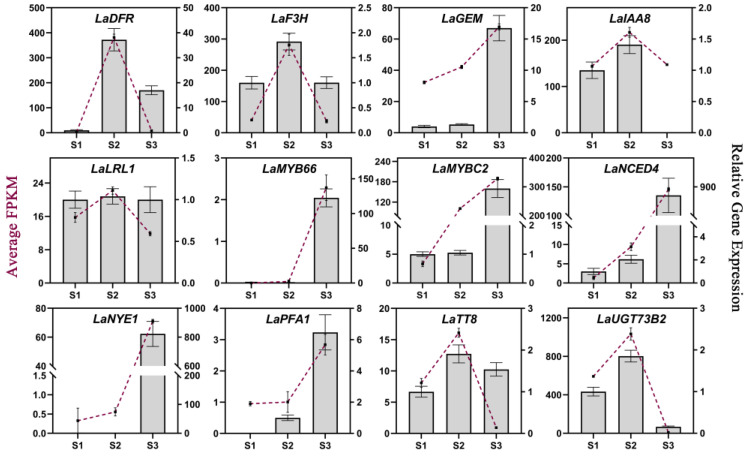
qRT-PCR validation of 12 hub genes. The average FPKMs obtained from RNA-seq analysis are shown as purple lines, corresponding to the left *Y*-axis. The relative gene expression levels obtained from qRT-PCR analysis are shown in histograms corresponding to the right *Y*-axis. β-actin was used as the internal control. The error bars represent the SDs of three biological replicates.

**Table 1 ijms-25-02436-t001:** The result of BUSCO assessment.

Class	Abbreviation	Proportion
Complete and single-copy	C,S	78.47%
Complete and duplicated	C,D	4.24%
Fragmented	F	5.07%
Missing	M	12.22%
Total	/	100.00%

**Table 2 ijms-25-02436-t002:** Important genes related to trichome development.

Gene Symbol in *Arabidopsis*	Gene Full Name	Regulation	Transcript ID
*GL1*	*GLABRA 1*	Positive	TRINITY_DN13618_c0_g2_i2_2
*TTG1*	*TRANSPARENT TESTA GLABRA 1*	Positive	TRINITY_DN15065_c0_g1_i1_1
*GL3*	*GLABRA 3*	Positive	TRINITY_DN15165_c1_g2_i2_1
*GL2*	*GLABRA 2*	Positive	TRINITY_DN11999_c0_g1_i1_3
*TTG2*	*TRANSPARENT TESTA GLABRA 2*	Positive	TRINITY_DN21499_c0_g1_i4_1
*EGL3*	*ENHANCER OF GLABRA 3*	Positive	TRINITY_DN21331_c0_g1_i1_1
*MYB23*	*MYB23*	Positive	TRINITY_DN13618_c0_g2_i2_2
*GIS*	*GLABROUS INFLORESCENCE STEMS*	Positive	TRINITY_DN64_c0_g1_i1_1
*GIS2*	*GLABROUS INFLORESCENCE STEMS 2*	Positive	TRINITY_DN6774_c0_g1_i1_3
*RHL2*	*ROOT HAIRLESS 2*	Positive	TRINITY_DN28312_c0_g1_i1_4
*STI*	*STICHEL*	Positive	TRINITY_DN18577_c0_g1_i1_1
*KLK*	*KLUNKER*	Positive	TRINITY_DN7541_c0_g1_i2_4
*BRK1*	*BRICK1*	Positive	TRINITY_DN7931_c0_g1_i1_2
*DIS1*	*DISTORTED TRICHOMES 1*	Positive	TRINITY_DN15188_c0_g1_i1_2
*DIS2*	*DISTORTED TRICHOMES 2*	Positive	TRINITY_DN15188_c0_g1_i1_2
*SPL9*	*SQUAMOSA PROMOTER BINDING PROTEIN-LIKE 9*	Positive	TRINITY_DN3340_c0_g1_i1_2
*TEM*	*TEMPRANILLO*	Positive	TRINITY_DN17099_c1_g1_i1_2
*TCS1*	*TEMPRANILLO*	Positive	TRINITY_DN19044_c0_g1_i1_2
*TT8*	*TRANSPARENT TESTA 8*	Positive	TRINITY_DN21331_c0_g1_i1_1
*MYC1*	*MYC1*	Positive	TRINITY_DN15165_c1_g2_i2_1
*TRY*	*TRIPTYCHON*	Negative	TRINITY_DN16344_c0_g3_i1_2
*CPC*	*CAPRICE*	Negative	TRINITY_DN16490_c0_g1_i1_3
*MYB66*	*MYB66*	Negative	TRINITY_DN13049_c0_g1_i1_2
*ETC1*	*ENHANCER OF TRY AND CPC 1*	Negative	TRINITY_DN2318_c0_g1_i1_4
*ETC2*	*ENHANCER OF TRY AND CPC 2*	Negative	TRINITY_DN2318_c0_g1_i1_4
*ETC3*	*ENHANCER OF TRY AND CPC 3*	Negative	TRINITY_DN2318_c0_g1_i1_4
*KAK*	*KAKTUS*	Negative	TRINITY_DN4349_c0_g1_i1_5

## Data Availability

The data from the transcriptome analysis can be found in the Appendix A.

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
