# Peer review of "Transcriptome Analysis Reveals Coexpression Networks and Hub Genes Involved in Papillae Development in Lilium auratum"

_ijms, 2024, doi:10.3390/ijms25042436_

Round 1
Reviewer 1 Report
Comments and Suggestions for Authors
The current study provides a nice comprehensive overview of the developmental pathway responsible for papillae formation in Lilium. The manuscript is nicely written and the analysis is portrayed beautifully. I have few comments especially comment 7 and comment 8 which must be addressed.
1. The abstract requires more clarity. terms like WGCNA and MEbrown needs to be introduced first briefly.
2. The results section should begin with the different developmental stages explaining the phenotypic differences. So Figure 7 should appropriately be Figure 1. It should also include the rationale for selecting these stages for analysis.
3. In Figure 1, panel b and d can be shifted as supplemental.
4. In figure 2, the transcript id should reflect the gene homolog name. It is tough to comprehend and comment on the expression of the genes.
5. For the coexpression network visualization in figure 4, I would suggest incorporating colour based on fold change for each node.
6. in the methods section the RNA sample collection and isolation should be clearly elaborated.
7. The expression profile of the hub genes in different developmental stages shown in Figure 6 can be studied in other cultivars with different floral colouration and papillae structures. This would give a clearer view of the importance of these hub genes in imparting colouration in Lilium species.
8. In the discussion section, the authors discuss of genes the expression of which has not been shown in figure 6. I would suggest the inclusion of these genes such as STAYGREEN in figure 6.
Comments on the Quality of English LanguageNone
Reviewer 2 Report
Comments and Suggestions for Authors
Dear authors,
Thank you and congratulations for the beautifully worked out data and analysis. The study have been conducted well and well drafted, thus, i have very few queries.
Line 95 needs a reference.
Can k means clustering to show the relatedness be added to fig 2 heatmaps? and also fig 3d.
Please add gene names for ease of the readers in table 1.
Stats are missing in fig 6.
Reviewer 3 Report
Comments and Suggestions for Authors
Authors performed transcriptome analysis of the genes accompanied by papillae developmnet in lilium auratum. The text is well written, while numerous corrections still require.
Summary:
There are too much abbrevaitions without explanation. I am not sure that average reader can understand so much abbreviations. Line 18: sequncuing of tepals can give you correlative data, not mechanism. Fort he mechanism you need to study in situ order of gene expression changes and order of the physiological process.
Line 49: colour is not importnat for plant development itself.
Line 51: “L.species” ¿
Line 53: “In addition to colour variation” – do you mean in flowers?
Line 56: enzyme activities, not enzyme itself.
Line 67: it will be greta to add some link ea. https://bmcplantbiol.biomedcentral.com/articles/10.1186/s12870-021-02840-x
Line 69: not in but out of epidermal tissue.
Line 73: trichomes itself does not have economic value. Grandular trichomes in tomato, for exmaple, secreted aromat and thia aromat have a value. But not trichomes itself!
Lines 76-77: “development of plant trichomes is regulated by a complex molecular
network, as well as phytohormones and environmental factors.” ¿?? Phytohormones induced molecular network.
Line 97 is quite similar to line 37.
Line 113: it is fine, but reader first need to go to the lines 328-240 to know what means S1, S2 etc. You can write as” stages 1 , 2.. (S1, S2..) for details see M&M.
Lines 114 – 122: It will be great to shown numerous numbers as small table for clarity.
Line 124: DEG análisis is not a role. You just find correlation between gene expression and morphology. If you want to tell about role, you need to artifically edit each of gene (CRISPR or other methods) and study effect of up or down expression of each of 5702 gene.
These data providfe you the role. So far it is only description of accompanied pathways. Fort he mechanism you need to find which promary signal (I experct hormonal one) orchestarted expression of al lof 5702 gene. And role of chromatin organization/epigenetic as posible intermediator of hormonal signal in this orchestra.
Figure 1: in the cirrent pdf resolution is quite low. It may happens during pdf formation of low dpi in orignal image. I think it is easy to repair.
Line 253: for the role, you need to créate mutant with these genes and study effcet of down-regulation of certian gene expression.
Lines 300.308: very good análisis. It will be nice to point out role of JAZ in auxin distribution.
Line 421: “the mechanism of action of lily papillae”¿??
Comments on the Quality of English Language
Some polishing are require
Round 2
Reviewer 1 Report
Comments and Suggestions for Authors
I am happy with the modifications made however some concerns still remain, which have been discussed as follows:
In Figure 2, unlike the expression of PAL the DFR, ANR, ANS and UFGT mostly occur in S2 wherein the phenotypic difference is minimum. A proper elaboration in this context would be helpful in the text.
In a manuscript, figures are numbered according to the position where it is cited. The study begins with the different developmental stages. You may include Figure 7 as a sub-figure 1a to avoid this confusion.
I am surprised that the NCBI submission ID was deleted. It is ethical to upload the data to a public repository before it is sent for review.
Reviewer 3 Report
Comments and Suggestions for Authors
The text is OK. Thank you for corrections!
Author Response
Dear Reviewer:
We would like to thank you for your careful reading, helpful comments, and constructive suggestions which have significantly improved the presentation of our manuscript. We really appreciate all your comments and suggestions! The quality and rigor of our manuscript continues to improve with the help of the Reviewer. We thank the Reviewer for the thoughtful review and comments on our manuscript, and we hope that the revised manuscript is suitable for publication in the International Journal of Molecular Sciences.
Sincerely,
Yuntao Zhu